# Reconstructing noisy gene regulation dynamics using extrinsic-noise-driven neural stochastic differential equations

**Jiancheng Zhang**[1☯], **Xiangting Li**[2☯], **Xiaolu Guo** [3☯*], **Zhaoyi You**[4], **Lucas Böttcher**[5,6], **Alex Mogilner**[7], **Alexander Hoffmann**[3], **Tom Chou**[2,8*], **Mingtao Xia**[9*]

**1** Department of Electrical and Computer Engineering, University of California, Riverside, California, United States of America, **2** Department of Computational Medicine, University of California, Los Angeles, California, United States of America, **3** Department of Microbiology, Immunology, and Molecular Genetics (MIMG) and Institute for Quantitative and Computational Biosciences, University of California, Los Angeles, California, United States of America, **4** Ray and Stephanie Lane Computational Biology Department, School of Computer Science, Carnegie Mellon University, Pittsburgh, Pennsylvania, United States of America, **5** Department of Computational Science and Philosophy, Frankfurt School of Finance and Management, Frankfurt am Main, Germany, **6** Laboratory for Systems Medicine, Department of Medicine, University of Florida, Gainesville, Florida, United States of America, **7** Courant Institute of Mathematical Sciences, New York University, New York, New York, United States of America, **8** Department of Mathematics, University of California, Los Angeles, California, United States of America, **9** Department of Mathematics, University of Houston, Houston, Texas, United States of America

☯ These authors contributed equally to this work.
* xiaoluguo@g.ucla.edu (XG); tomchou@ucla.edu (TC); mxia4@uh.edu (MX)

**Data availability statement:** No data was created in this research. All data used in this

## Abstract

Proper regulation of cell signaling and gene expression is crucial for maintaining cellular function, development, and adaptation to environmental changes. Reaction dynamics in cell populations is often noisy because of (i) inherent stochasticity of intracellular biochemical reactions ("intrinsic noise") and (ii) heterogeneity of cellular states across different cells that are influenced by external factors ("extrinsic noise"). In this work, we introduce an extrinsic-noise-driven neural stochastic differential equation (END-nSDE) framework that utilizes the Wasserstein distance to accurately reconstruct SDEs from stochastic trajectories measured across a heterogeneous population of cells (extrinsic noise). We demonstrate the effectiveness of our approach using both simulated and experimental data from three different systems in cell biology: (i) circadian rhythms, (ii) RPA-DNA binding dynamics, and (iii) NFκB signaling processes. Our END-nSDE reconstruction method can model how cellular heterogeneity (extrinsic noise) modulates reaction dynamics in the presence of intrinsic noise. It also outperforms existing time-series analysis methods such as recurrent neural networks (RNNs) and long short-term memory networks (LSTMs). By inferring cellular heterogeneities from data, our END-nSDE reconstruction method can reproduce noisy dynamics observed in experiments. In summary, the reconstruction method we propose offers a useful surrogate modeling approach for complex biophysical processes, where high-fidelity mechanistic models may be impractical.

research are publicly available at https://www.nature.com/articles/s41467-023-39579-y and https://www.embopress.org/doi/full/10.1038/s44320-024-00047-4 and have been properly cited. The simulated datasets, neural SDE model code, and analysis scripts to replicate the study findings are available on GitHub at https://github.com/JianchengZ/Neural-SDE-GeneDynamics.

**Funding:** XG acknowledges financial support from UCLA Collaboratory Fellowship. LB acknowledges financial support from hessian.AI and the ARO through grant W911NF-23-1-0129. TC acknowledges inspiring discussions at the "Statistical Physics and Adaptive Immunity" program at the Aspen Center for Physics, which is supported by the National Science Foundation grant PHY-2210452. The funders had no role in study design, data collection and analysis, decision to publish, or preparation of the manuscript. XG and AH acknowledge support from NIH R01AI173214.

**Competing interests:** The authors have declared that no competing interests exist.

## Author summary

In this work, we propose extrinsic-noise-driven neural stochastic differential equations (END-nSDE) to reconstruct noisy regulated gene expression dynamics. One of our main contributions is that we generalize a recent Wasserstein-distance-based SDE reconstruction approach to incorporate extrinsic noise (parameters that vary across different cells). Our approach can thus capture intrinsic fluctuations in gene regulatory dynamics driven by extrinsic noise (heterogeneity among cells), offering an advantage over deterministic models and outperforming other benchmarks. By inferring noise intensities from batches of experimental data, our END-nSDE can partially capture experimental noisy signaling dynamic data and provides a surrogate model for biomolecular processes that are too complex to model directly.

## 1. Introduction

Reactions that control signaling and gene regulation are important for maintaining cellular function, development, and adaptation to environmental changes, which impact all aspects of biological systems, from embryonic development to an organism's ability to sense and respond to environmental signals. Variations in gene regulation, arising from noisy biochemical processes [1,2], can result in phenotypic heterogeneity even in a population of genetically identical cells [3].

Noise within cell populations can be categorized as (i) "intrinsic noise," which arises from the inherent stochasticity of biochemical reactions and quantifies, *e.g.*, biological variability across cells in the same state [2,4,5], and (ii) "extrinsic noise," which encompasses heterogeneities in environmental factors or differences in cell state across a population. A substantial body of literature has focused on quantifying intrinsic and extrinsic noise from experimental and statistical perspectives [1,2,6–13]. Experimental studies have specifically identified relevant sources of noise in various organisms, including *E. coli* (Escherichia coli), yeast, and mammalian systems [2,14–17].

Extrinsic noise is associated with uncertainties in biological parameters that vary across different cells. The distribution over physical and chemical parameters determine the observed variations in cell states, concentrations, locations of regulatory proteins and polymerases [1,2,18], and transcription and translation rates [19]. For example, extrinsic noise is the main contributor to the variability of concentrations of oscillating p53 protein levels across cell populations [20]. On the other hand, intrinsic noise, *i.e.*, inherent stochasticity of cells in the same state, can limit the accuracy of expression and signal transmission [2,5]. Based on the law of mass action [21,22], ordinary differential equations (ODEs) apply only in some deterministic or averaged limit and do not take into account intrinsic noise. Therefore, stochastic models are necessary to accurately represent biological processes, such as thermodynamic fluctuations inherent to molecular interactions within regulatory networks [1,5,18] or random event times in birth-death processes.

Existing stochastic modeling methods that account for intrinsic noise include Markov jump processes [23,24] and SDEs [25–27]. These approaches are applicable to different system sizes: Markov jump processes provide exact descriptions for discrete molecular systems, while SDEs serve as continuous approximations to Markov processes when molecular abundances are sufficiently high. SDE approaches may not be suitable for gene expression systems with very low copy numbers, where discrete master equation descriptions are more accurate. However, SDE approaches become more appropriate when modeling protein dynamics or when

gene regulatory interactions are modeled implicitly through Hill functions. Additionally, a hierarchical Markov model was designed in [28] for parameter inference in dual-reporter experiments to separate the contributions of extrinsic noise, intrinsic noise, and measurement error when both extrinsic and intrinsic noise are present. The described methods have been effective in the reconstruction of low-dimensional noisy biological systems. Discrete master-equation methods to model the evolution of probabilities in systems characterizing, *e.g.*, gene regulatory dynamics [29–31], can be computationally expensive and usually require specific forms of a stochastic model with unknown parameters that need to be inferred. It is unclear whether such methods and their generalizations can be applied to more complex (*e.g.*, higher-dimensional) systems for which a mechanistic description of the underlying biophysical dynamics is not available or impractical.

SDEs can capture both the mean dynamics (as ODEs do) and random fluctuations, offering a practical and scalable alternative to master equations in complex systems. Thus, we introduce an extrinsic-noise-driven neural stochastic differential equation (END-nSDE) reconstruction method that builds upon a recently developed Wasserstein distance ($W_2$ distance) nSDE reconstruction method [32]. Our method is used to identify macromolecular reaction kinetics and cell signaling dynamics from noisy observational data in the presence of *both extrinsic and intrinsic noise*. A key question we address in this paper is how extrinsic noise that characterizes cellular heterogeneity influences the overall stochastic dynamics of the population.

The major differences between the approach presented here and prior work [32] are: (i) the inclusion of extrinsic noise into the framework allowing one to model cell-to-cell variability through parameter heterogeneity, and (ii) the ability of our method to learn the dependency of the SDE on those parameters, enabling reconstruction of a family of SDEs rather than a single SDE model. In contrast, the method developed in reference [32] focuses on reconstructing a single SDE without considering parameter variations or extrinsic noise sources. In Fig 1, we provide an overview of the specific applications that we study in this work.

Our approach employs neural networks as SDE approximators in conjunction with the `torchsde` package [33,34] for reconstructing noisy dynamics from data. Previous work showed that for SDE reconstruction tasks, the $W_2$ distance nSDE reconstruction method outperforms other benchmark methods such as generative adversarial networks [32,35]. Compared to other probabilistic metrics such as the KL divergence, the Wasserstein distance better incorporates the metric structure of the underlying space. This geometric property makes the Wasserstein distance particularly suitable for trajectory and image data on high-dimensional manifolds, where the supports of different distributions do not always overlap [36]. Additionally, the $W_2$-distance-based nSDE reconstruction method can directly extract the underlying SDE from temporal trajectories without requiring specific mathematical forms of the terms in the underlying SDE model. We apply our END-nSDE methodology to three biological processes: (i) circadian clocks, (ii) RPA-DNA binding dynamics, and (iii) NFκB signaling to illustrate the effectiveness of the END-nSDE method in predicting how extrinsic noise modulates stochastic dynamics with intrinsic noise. Additionally, our method demonstrates superior performance compared to several time-series modeling methods including recurrent neural networks (RNNs), long short-term memory networks (LSTMs), and Gaussian processes. In summary, the reconstruction method we propose provides a useful surrogate modeling approach for complex biophysical and biochemical processes, especially in scenarios where high-fidelity mechanistic models are impractical.

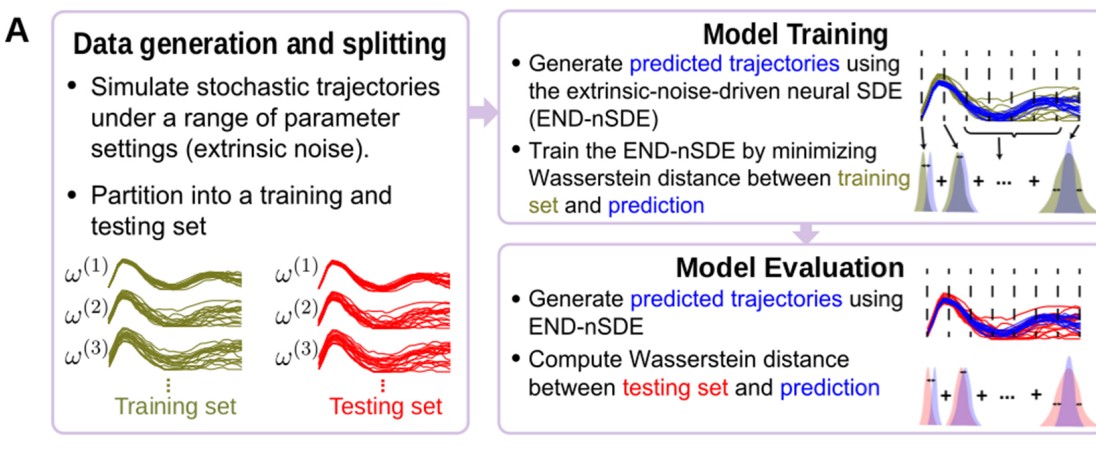

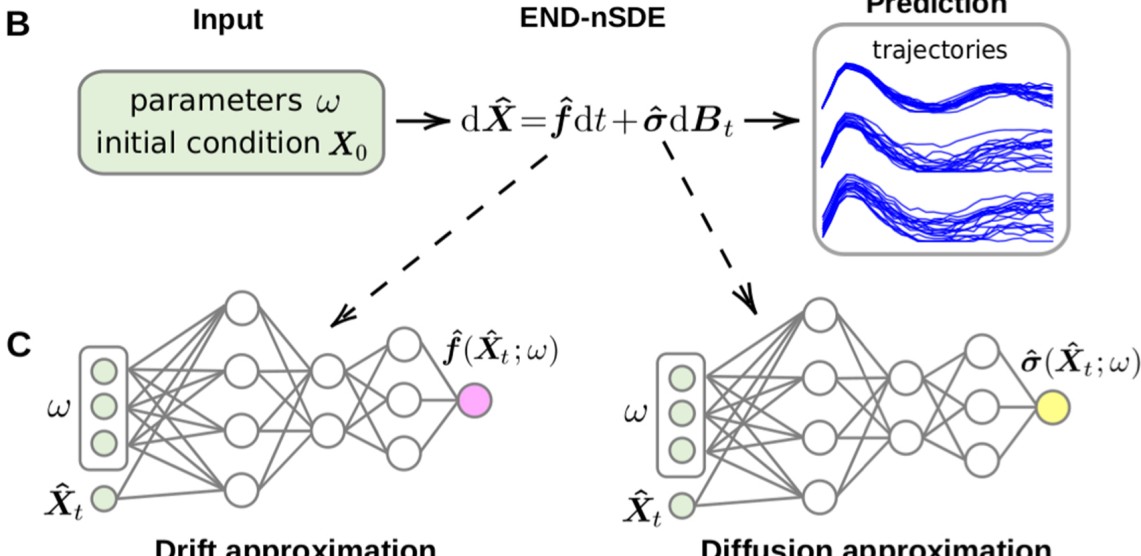

**Fig 1. Workflow of our proposed END-nSDE prediction on parameters altering stochastic dynamics.** A. Workflow for training and testing of the extrinsic-noise-driven neural SDE (END-nSDE). Predicted trajectories are simulated (see B) using a range of model parameters (see Sect 2.2) before splitting into training and testing sets (see Fig E in S1 Text for details on the splitting strategy). Model parameters and state variables serve as inputs to a neural network that reconstructs drift and diffusion terms (see C). Network weights are optimized by minimizing the Wasserstein distance (Eq 8) between the training set and predicted trajectories. B. Predicted trajectories are generated by the reconstructed SDE $d\hat{X} = \hat{f}(\hat{X};\omega)dt + \hat{\sigma}(\hat{X};\omega)dB_t$. C. The drift and diffusion functions, $\hat{f}$ and $\hat{\sigma}$, are

approximated using parameterized neural networks. The parameterized neural-network-based drift function $\hat{f}(\hat{X}; \omega)$ and diffusion function $\hat{\sigma}(\hat{X}; \omega)$ take the system state $\hat{X}$ and biological parameters $\omega$ as inputs. D. Table of three examples illustrating the nSDE input, along with training and testing datasets. For the last, NF$\kappa$B example, a more detailed workflow for validation on experimental datasets is illustrated in Fig 8.

## 2. Methods and models

In this work, we extend the temporally decoupled squared $W_2$-distance SDE reconstruction method proposed in Refs. [32,37] to reconstruct noisy dynamics across a heterogeneous cell population ("extrinsic noise"). Our goal is to not only reconstruct SDEs for approximating noisy cellular signaling dynamics from time-series experimental data, but to also quantify how heterogeneous biological parameters, such as enzyme- or kinase-mediated biochemical reaction rates, affect such noisy cellular signaling dynamics.

### 2.1. SDE reconstruction with heterogeneities in biological parameters

The $W_2$-distance-based neural SDE reconstruction method proposed in [32] aims to approximate the SDE

$$d\boldsymbol{X}(t) = \boldsymbol{f}(\boldsymbol{X}(t), t)dt + \boldsymbol{\sigma}(\boldsymbol{X}(t), t)d\mathbf{B}(t), \ \boldsymbol{X}(t) \in \mathbb{R}^d, \tag{1}$$

using an approximated SDE

$$d\hat{\boldsymbol{X}}(t) = \hat{\boldsymbol{f}}(\hat{\boldsymbol{X}}(t), t)dt + \hat{\boldsymbol{\sigma}}(\hat{\boldsymbol{X}}(t), t)d\mathbf{B}(t), \ \hat{\boldsymbol{X}}(t) \in \mathbb{R}^d, \tag{2}$$

where $\hat{\boldsymbol{f}}$ and $\hat{\boldsymbol{\sigma}}$ are two parameterized neural networks that approximate the drift and diffusion functions $\boldsymbol{f}$ and $\boldsymbol{\sigma}$ in Eq (1), respectively. These two neural networks are trained by minimizing a temporally decoupled squared $W_2$-distance loss function

$$\tilde{W}_2^2(\mu, \hat{\mu}) = \int_0^T \inf_{\pi \in \Pi(\mu(t), \hat{\mu}(t))} \mathbb{E}_\pi \left[ \left\| \boldsymbol{X}(t) - \hat{\boldsymbol{X}}(t) \right\|^2 \right] dt, \tag{3}$$

where $\Pi(\mu(t), \hat{\mu}(t))$ denotes the set of all coupling distributions $\pi$ of two distributions $\mu(t), \hat{\mu}(t)$ on the probability space $\mathbb{R}^d$, and $\boldsymbol{X}(t)$ and $\hat{\boldsymbol{X}}(t)$ are the observed trajectories at time $t$ and trajectories generated by the approximate SDE model Eq (2) at time $t$, respectively. $\mu$ and $\hat{\mu}$ are the probability distributions associated with the stochastic processes $\{\boldsymbol{X}(t)\}, 0 \le t \le T$ and $\{\hat{\boldsymbol{X}}(t)\}, 0 \le t \le T$, respectively, while $\mu(t)$ and $\hat{\mu}(t)$ are the probability distributions of $\boldsymbol{X}(t)$ and $\hat{\boldsymbol{X}}(t)$ at a specific time $t$. A coupling $\pi \in \Pi(\mu(t), \hat{\mu}(t))$ between $\mu(t)$ and $\hat{\mu}(t)$ is defined by

$$\pi(A, \mathbb{R}^d) = \mu(A), \ \pi(\mathbb{R}^d, B) = \hat{\mu}(B), \ \forall A, B \in \mathcal{B}(\mathbb{R}^d), \tag{4}$$

where $\mathcal{B}(\mathbb{R}^d)$ is the Borel $\sigma$-algebra on $\mathbb{R}^d$ and $\mathbb{E}_\pi \left[ \left\| \boldsymbol{X}(t) - \hat{\boldsymbol{X}}(t) \right\|^2 \right]$ represents the expectation when $(\boldsymbol{X}, \hat{\boldsymbol{X}}) \sim \pi$.

The $\tilde{W}_2^2$ term in Eq (3) is denoted as the temporally decoupled squared $W_2$ distance loss function. For simplicity, in this paper, we shall also denote Eq (3) as the squared $W_2$ loss. The infimum is taken over all possible coupling distributions $\pi \in \Pi(\mu(t), \hat{\mu}(t))$ and $\| \cdot \|$ denotes the $\ell^2$ norm of a vector. That is,

$$\left\| \boldsymbol{X}(t) \right\|^2 := \sum_{i=1}^d \left| X_i(t) \right|^2. \tag{5}$$

Across different cells, extrinsic noise or cellular heterogeneities such as differences in kinase or enzyme abundances resulting from cellular variabilities, can lead to variable, cell-specific, gene regulatory dynamics. Such heterogeneous and stochastic gene expression (both intrinsic and extrinsic noise) can be modeled using SDEs with distributions of parameter values reflecting cellular heterogeneity. To address heterogeneities in gene dynamics across different cells, we propose an END-nSDE method that is able to reconstruct a family of SDEs for the same gene expression process under different parameters. Specifically, for a given set of (biological) parameters $\omega$, we are interested in reconstructing

$$\mathrm{d}X(t;\omega) = f(X(t;\omega);\omega)\mathrm{d}t + \sigma(X(t;\omega);\omega)\mathrm{d}B(t), \tag{6}$$

using the approximate SDE

$$\mathrm{d}\hat{X}(t;\omega) = \hat{f}(\hat{X}(t;\omega);\omega)\mathrm{d}t + \hat{\sigma}(\hat{X}(t;\omega);\omega)\mathrm{d}\hat{B}(t), \tag{7}$$

in the sense that the errors $f(X(t;\omega);\omega) - \hat{f}(X(t;\omega);\omega)$ and $\sigma(X(t;\omega);\omega) - \hat{\sigma}(X(t;\omega);\omega)$ for all different values of $\omega$ will be minimized. In Eq (7), $\hat{f}$ and $\hat{\sigma}$ are represented by two parameterized neural networks that take both the state variable $\hat{X}$ and the parameters $\omega$ as inputs. To train these two neural networks, we propose an extrinsic-noise-driven temporally decoupled squared $W_2$ distance loss function

$$L(\Lambda) = \sum_{\omega \in \Lambda} \tilde{W}_2^2(\mu(\omega), \hat{\mu}(\omega)), \tag{8}$$

where $\mu(\omega)$ and $\hat{\mu}(\omega)$ are the distributions of the trajectories $X(t;\omega)$, $0 \le t \le T$ and $\hat{X}(t;\omega)$, $0 \le t \le T$, and $\tilde{W}$ is the temporally decoupled squared $W_2$ loss function in Eq (3). $\Lambda$ denotes the set of parameters $\omega$. Eq (8) is different from the local squared $W_2$ loss in Refs. [38,39] since we do not require a continuous dependence of $\{X(t;\omega)\}_{t\in[0,T]}$ on the parameter $\omega$ nor do we require that $\omega$ is a continuous variable. The extrinsic-noise-driven temporally decoupled squared $W_2$ loss function Eq (8) takes into account both parameter heterogeneity and intrinsic fluctuations as a result of the Wiener processes $B(t)$ and $\hat{B}(t)$ in Eqs (1) and (2).

Our END-nSDE method is outlined in Fig 1A–1C. With observed noisy single-cell dynamic trajectories as the training data, we train two parameterized neural networks [40] by minimizing Eq (8) to approximate the drift and diffusion terms in the SDE. The reconstructed nSDE is a surrogate model of single-cell dynamics (see Fig 1B and 1C). The hyperparameters and settings for training the neural SDE model are summarized in Table A in S1 Text. Through the examples outlined in Fig 1D, we will show that our $W_2$-distance-based method can yield very small errors in the reconstructed drift and diffusion functions $f - \hat{f}$ and $\sigma - \hat{\sigma}$.

## 2.2. Biological models

We consider three biological examples where stochastic dynamics play a critical role and use our END-nSDE method to reconstruct noisy single-cell gene expression dynamics under both intrinsic and extrinsic noise (also summarized in Fig 1D). In these applications, we investigate the extent to which the END-nSDE can efficiently capture and infer changes in the dynamics driven by extrinsic noise.

**2.2.1. Noisy oscillatory circadian clock model.**  Circadian clocks, often with a typical period of approximately 24 hours, are ubiquitous in intrinsically noisy biological rhythms generated at the single-cell molecular level [41].

**Algorithm 1 END-nSDE training and prediction framework.**

```
Obtain training trajectories {X(t;ω)} (simulated or experimental
time-series data). Maximum training epochs = i_max.
Preprocess the relevant training trajectories by grouping them
according to different biophysical parameters ω.
```
**Phase 1: Training**
**for** $i \leq i_{max}$ **do**
```
   Input the initial state X₀ and ω ∈ Λ into the END-nSDE to generate
   new predictions X̂(t;ω).
   Calculate the loss function L(Λ) in Eq (8) and perform gradient
descent to train the END-nSDE model.
```
**end for**
**return** the trained END-nSDE model
**Phase 2: Prediction**
```
Input initial condition X₀ and corresponding noise parameters ω from
testing data into the trained END-nSDE model.
Generate predicted trajectories X̂(t;ω) from the learned model.
```

We consider a minimal SDE model of the periodic gene dynamics responsible for *per* gene expression which is critical in the circadian cycle. Since *per* gene expression is subject to intrinsic noise [42], we describe it using a linear damped-oscillator SDE

$$
\begin{aligned}
\mathrm{d}x &= -\alpha x \mathrm{d}t - \beta y \mathrm{d}t + \xi_{x1}\mathrm{d}B_{1,t} + \xi_{x2}\mathrm{d}B_{2,t} \\
\mathrm{d}y &= \beta x \mathrm{d}t - \alpha y \mathrm{d}t + \xi_{y1}\mathrm{d}B_{1,t} + \xi_{y2}\mathrm{d}B_{2,t},
\end{aligned}
\tag{9}
$$

where $x$ and $y$ are the dimensionless concentrations of the *per* mRNA transcript and the corresponding *per* protein, respectively. $\mathrm{d}B_{1,t}$, $\mathrm{d}B_{2,t}$ are two independent Wiener processes and the parameters $\alpha > 0$ and $\beta > 0$ denote the damping rate and angular frequency, respectively. A stability analysis at the steady state $(x, y) = (0, 0)$ in the noise-free case ($\xi_x = \xi_y = 0$ in Eq (9)) reveals that the real parts of the eigenvalues of the Jacobian matrix $\begin{pmatrix} -\alpha & -\beta \\ \beta & -\alpha \end{pmatrix}$ at $(x, y) = (0, 0)$ are all negative, indicating that the origin is a stable steady state when the system is noise-free. Noise prevents the state $(x(t), y(t))$ from remaining at $(0,0)$; thus, fluctuations in the single-cell circadian rhythm are noise-induced [42].

To showcase the effectiveness of our proposed END-nSDE method, we take different forms of the diffusion functions $\xi_x$ and $\xi_y$ in Eq (9), accompanied by different values of noise strength and the correlation between the diffusion functions in the dynamics of $x, y$.

**2.2.2. RPA-DNA binding model.** Regulation of gene expression relies on complex interactions between proteins and DNA, often described by the kinetics of binding and dissociation. Replication protein A (RPA) plays a pivotal role in various DNA metabolic pathways, including DNA replication and repair, through its dynamic binding with single-stranded DNA (ssDNA) [43–46]. By modulating the accessibility of ssDNA, RPA regulates multiple biological mechanisms and functions, acting as a critical regulator within the cell [47]. Understanding the dynamics of RPA-ssDNA binding is therefore a research area of considerable biological interest and significance.

Multiple binding modes and volume exclusion effects complicate the modeling of RPA-ssDNA dynamics. The RPA first binds to ssDNA in 20 nucleotide (nt) mode, which occupies 20nt of the ssDNA. When the subsequent 10nt of ssDNA is free, 20nt-mode RPA can transform to 30nt-mode, further stabilizing its binding to ssDNA, as illustrated in Fig 2. Occupied ssDNA is not available for other proteins to bind. Consequently, the gap size between adjacent ssDNA-bound RPAs determines the ssDNA accessibility to other proteins.

Mean-field mass-action type chemical kinetic ODE models cannot describe the process very well because they do not capture the intrinsic stochasticity. A stochastic model that

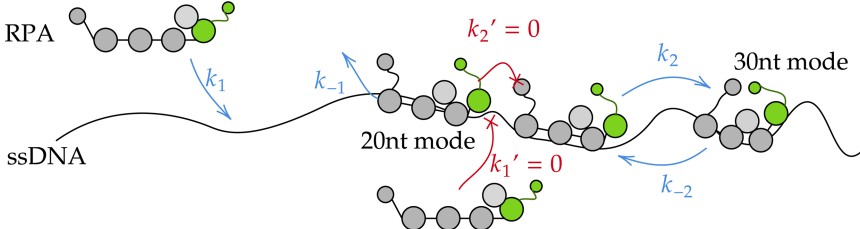

**Fig 2. A continuous-time discrete Markov chain model for multiple RPA molecules binding to long ssDNA.** The possible steps in the biomolecular kinetics of multiple RPA molecules binding to ssDNA. The RPA in the free solution can bind to ssDNA with rate $k_1$ provided there are at least 20 nucleotides (nt) of consecutive unoccupied sites. This bound "20nt mode" RPA unbinds with rate $k_{-1}$. When space permits, the 20nt-mode RPA can extend and bind an additional 10nt of DNA at a rate of $k_2$, converting it to a 30nt-mode bound protein. The 30nt-mode RPA transforms back to 20nt-mode spontaneously with the rate $k_{-2}$. However, when the gap is not large enough to accommodate the RPA, the binding or conversion is prohibited ($k_1' = 0$ and $k_2' = 0$).

tracks the fraction of two different binding modes of RPA, 20nt-mode ($x_1$) and 30nt-mode ($x_2$), has been developed to capture the dynamics of this process. A brute-force approach using forward stochastic simulation algorithms (SSAs) [48] was then used to fit the model to experimental data [47]. However, a key challenge in this approach is that the model is nondifferentiable with respect to the kinetic parameters, making it difficult to estimate parameters. Yet, simple spatially homogeneous stochastic chemical reaction systems can be well approximated by a corresponding SDE of the form given in Eq (1) when the variables are properly scaled in the large system size limit [49]. While interparticle interactions shown in Fig 2 make it difficult to find a closed-form SDE approximation, results from [49] motivate the possibility of an SDE approximation for the RPA-ssDNA binding model in terms of the variables $x_1$ and $x_2$.

Here, to address the non-differentiability issue associated with the underlying Markov process, we use our END-nSDE model to construct a differentiable surrogate for SSAs, allowing it to be readily trained from data. Further details on the models and data used in this study are provided in Appendix B of S1 Text. Throughout our analysis of RPA-DNA binding dynamics, we benchmark the SDE reconstructed by our extended $W_2$-distance approach against those found using other time series analysis and reconstruction methods such as the Gaussian process, RNN, LSTM, and the neural ODE model. We show that our surrogate SDE model is most suitable for approximating the RPA-DNA binding process because it can capture the intrinsic stochasticity in the dynamics.

**2.2.3. NF$\kappa$B signaling model.** Macrophages can sense environmental information and respond accordingly with stimulus-response specificity encoded in signaling pathways and decoded by downstream gene expression profiles [50]. The temporal dynamics of NF$\kappa$B, a key transcription factor in immune response and inflammation, encodes stimulus information [51]. NF$\kappa$B targets and regulates vast immune-related genes [52–54]. While NF$\kappa$B signaling dynamics are stimulus-specific, they exhibit significant heterogeneity across individual cells under identical conditions [51]. Understanding how specific cellular heterogeneity (extrinsic noise) contributes to heterogeneity in NF$\kappa$B signaling dynamics can provide insight into how noise affects the fidelity of signal transduction in immune cells.

A previous modeling approach employs a 52-dimensional ODE system to quantify the NF$\kappa$B signaling network [51] and recapitulate the signaling dynamics of a representative cell. This ODE model includes 52 molecular entities and 47 reactions across a TNF-receptor module, an adaptor module, and a core module with and NF$\kappa$B-IKK-I$\kappa$B$\alpha$ (I$\kappa$B$\alpha$ is an inhibitor

of NFκB, while IKK is the IκB kinase complex that regulates the IκBα degradation) feedback loop (see Fig 3) [55]. However, such an ODE model is deterministic and assumes no intrinsic fluctuations in the biomolecular processes. Yet, from experimental data, the NFκB signaling dynamics fluctuate strongly; such fluctuations cannot be quantitatively described by any deterministic ODE model. Due to the system's high dimensionality and nonlinearity, it is challenging to quantify how intrinsic noise influences temporal coding in NFκB dynamics.

To incorporate the intrinsic noise within the NFκB signaling network, we introduce noise terms into the 52-dimensional ODE system to build an SDE that can account for the observed temporally fluctuating nuclear NFκB trajectories. While NFκB signaling pathways involve many variables, experimental constraints limit the number of measurable components. Among these, nuclear NFκB concentration is the most direct and critical experimental readout. As a minimal stochastic model, we hypothesize that only the biophysical and biochemical processes of NFκB translocation (which directly affects experimental measurements) and IκBα transcription (a key regulator of NFκB translocation) are subject to Brownian-type noise (red arrows in Fig 3), as these processes play crucial roles in the oscillatory dynamics of NFκB [55].

The intensity of Brownian-type noise in the NFκB dynamics may depend on factors such as cell volume (smaller volumes result in higher noise intensity), or copy number (lower copy numbers lead to greater noise intensity), and is therefore considered a form of extrinsic noise. Noise intensity parameters thus capture an aspect of cellular heterogeneity. There are other sources of cellular heterogeneity, such as variations in kinase or enzyme abundances, which

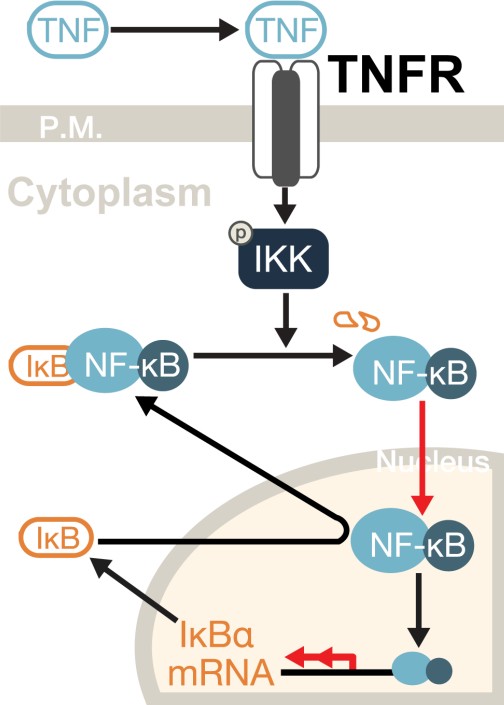

**Fig 3. Simplified schematic of the NFκB Signaling Network.** TNF binds its receptor, activating IKK, which degrades IκBα and releases NFκB. The free NFκB translocates to the nucleus and promotes IκBα transcription. Newly synthesized IκBα then binds NFκB and exports it back to the cytoplasm. Red arrows indicate noise that we consider in the corresponding SDE system.

are too complicated to model and are thus not included in the current model. For simplicity, all kinetic parameters, except for the noise intensity ($\sigma$), are assumed to be consistent with those of a representative cell [55]. The 52-dimensional ODE model for describing NF$\kappa$B dynamics is given in Refs. [51,56]. We extend this model by adding noise to the dynamics of the sixth, ninth, and tenth ODEs of the 52-dimensional ODE model. We retain 49 ODEs but convert the equations for the sixth, ninth, and tenth components to SDEs:

$$\mathrm{d}u_6 = \Big( k_{\mathrm{basal}} + \frac{k_{\mathrm{max}} u_{52}^{n_{\mathrm{NF\kappa B}}}}{u_{52}^{n_{\mathrm{NF\kappa B}}} + K_{\mathrm{NF\kappa B}}^{n_{\mathrm{NF\kappa B}}}} - k_{\mathrm{deg}} u_6 \Big)\mathrm{d}t + \sigma_1 \mathrm{d}B_{1,t}$$

$$\mathrm{d}u_9 = \big( k_{\mathrm{imp}} u_9 - k_{\mathrm{a\text{-}I\kappa B\text{-}NF\kappa B}} u_2 u_9 - k_{\mathrm{deg\text{-}NF\kappa B}} u_9 + v^{-1} k_{\mathrm{exp}} u_{10} + k_{\mathrm{d\text{-}I\kappa B\text{-}NF\kappa B}} u_4 + k_{\mathrm{phos}} u_7 \big)\mathrm{d}t - \sigma_2 \mathrm{d}B_{2,t}$$

$$\mathrm{d}u_{10} = \big( -k_{\mathrm{exp}} u_{10} - k_{\mathrm{a\text{-}I\kappa B\text{-}NF\kappa B}} u_3 u_{10} + v k_{\mathrm{imp}} u_9 + k_{\mathrm{d\text{-}I\kappa B\text{-}NF\kappa B}} u_5 \big)\mathrm{d}t + \sigma_2 \mathrm{d}B_{2,t}.$$

$$(10)$$

In Eqs 10, $u_2$ is the concentration of I$\kappa$B$\alpha$ in the cytoplasm; $u_3$ is the concentration of I$\kappa$B$\alpha$ in the nucleus; $u_4$ is the concentration of the I$\kappa$B$\alpha$-NF$\kappa$B complex; $u_5$ is the concentration of the I$\kappa$B$\alpha$-NF$\kappa$B complex in the nucleus; $u_6$ is the mRNA of I$\kappa$B$\alpha$; $u_7$ is the IKK-I$\kappa$B$\alpha$-NF$\kappa$B complex; $u_9$ is NF$\kappa$B; $u_{10}$ represents nuclear NF$\kappa$B concentration; and $u_{52}$ is the nuclear concentration of NF$\kappa$B with RNA polymerase II that is ready to initiate mRNA transcription. A description of the parameters and their typical values are given in Table C in S1 Text. The quantities $\sigma_1 \mathrm{d}B_{1,t}$ and $\sigma_2 \mathrm{d}B_{2,t}$ are noise terms associated with I$\kappa$B$\alpha$ transcription and NF$\kappa$B translocation, respectively. The remaining variables are latent variables and their dynamics are regulated via the remaining 49-dimensional ODE in Refs. [51,56]. The activation of NF$\kappa$B is quantified by the total nuclear NF$\kappa$B concentration ($u_5 + u_{10}$), which is also measured in experiments.

Within this example, we wish to determine if our proposed parameter-associated nSDE can accurately reconstruct the dynamics underlying experimentally observed NF$\kappa$B trajectory data.

## 3. Results

### 3.1. Accurate reconstruction of circadian clock dynamics

As an illustrative example, we use the $W_2$-distance nSDE reconstruction method to first reconstruct the minimal model for damped oscillatory circadian dynamics (see Eq (9)) under different forms of the diffusion function. We set the two parameters $\alpha$ = 0.19 and $\beta$ = 0.21 in Eq (9) and impose three different forms for the diffusion functions $\xi_{x1}, \xi_{x2}, \xi_{y1}, \xi_{y2}$: a constant diffusion function [57], a Langevin [58] diffusion function, and a linear diffusion function [59]. These functions, often used to describe fluctuating biophysical processes, are

$$\text{const:} \begin{bmatrix} \xi_{x1} & \xi_{x2} \\ \xi_{y1} & \xi_{y2} \end{bmatrix} = \sigma_0 \begin{bmatrix} 1 & c \\ c & 1 \end{bmatrix}, \tag{11}$$

$$\text{Langevin:} \begin{bmatrix} \xi_{x1} & \xi_{x2} \\ \xi_{y1} & \xi_{y2} \end{bmatrix} = \sigma_0 \begin{bmatrix} \sqrt{|x|} & c\sqrt{|y|} \\ c\sqrt{|x|} & \sqrt{|y|} \end{bmatrix}, \tag{12}$$

and

$$\text{linear:} \begin{bmatrix} \xi_{x1} & \xi_{x2} \\ \xi_{y1} & \xi_{y2} \end{bmatrix} = \sigma_0 \begin{bmatrix} x & c|y| \\ c|x| & y \end{bmatrix}. \tag{13}$$

There are two additional parameters in Eqs (11), (12), and (13): $\sigma_0$ that determines the intensity of the Brownian-type fluctuations and $c$ that controls the correlation of fluctuations

between the two dimensions. For each type of diffusion function, we trained a different nSDE model, each of which takes the state variables $(x,y)$ and the two parameters $(c, \sigma_0)$ as inputs and which outputs the values of the reconstructed drift and diffusion functions.

We take 25 combinations of $(\sigma_0, c) \in \{(0.1 + 0.05i, 0.2 + 0.2j), i \in \{0, ..., 4\}, j \in \{0, ..., 4\}\}$; for each combination of $(\xi_{x1}, \xi_{x2}, \xi_{y1}, \xi_{y2})$, we generate 50 trajectories from the ground truth SDE (9) as the training data with $t \in [0, 1]$. The initial condition is set as $(x(0), y(0)) = (0, 1)$. To test the accuracy of the reconstructed diffusion and drift functions, we measure the following relative errors:

$$\text{Error in } \boldsymbol{f} := \frac{\sum_{i=1}^{M} \sum_{j=0}^{T} |\boldsymbol{f}(\boldsymbol{X}_i(t_j; \omega); \omega) - \hat{\boldsymbol{f}}(\hat{\boldsymbol{X}}_i(t_j; \omega); \omega)|_1}{\sum_{j=0}^{T} |\boldsymbol{f}(\boldsymbol{X}_i(t_j; \omega); \omega)|_1}, \tag{14}$$

$$\text{Error in } \boldsymbol{\sigma} := \frac{\sum_{i=1}^{M} \sum_{j=0}^{T} \left\| \boldsymbol{\sigma}(\boldsymbol{X}_i(t_j; \omega); \omega) \boldsymbol{\sigma}^T(\boldsymbol{X}_i(t_j; \omega), t_j; \omega) \right| - \left| \hat{\boldsymbol{\sigma}}(\boldsymbol{X}_i(t_j; \omega); \omega) \hat{\boldsymbol{\sigma}}^T(t_j; \omega) \right\|_{\mathrm{m}}}{\sum_{i=0}^{M} \sum_{j=0}^{d} \left| \boldsymbol{\sigma}(\boldsymbol{X}_i(t_j; \omega); \omega) \boldsymbol{\sigma}^T(\boldsymbol{X}_i(t_j; \omega); \omega) \right|_{\mathrm{m}}}. \tag{15}$$

Here, $\boldsymbol{f} := (-\alpha x - \beta y, \beta x - \alpha y)^T$ is the vector of ground truth drift functions and $\hat{\boldsymbol{f}}$ is the reconstructed drift function. $\boldsymbol{\sigma}$ is the matrix of ground truth diffusion functions $[\xi_{x1}, \xi_{x2}; \xi_{y1}, \xi_{y2}]$ given in Eqs (11), (12), and (13). $M$ is the number of training samples, $|\cdot|_1$ denotes the $\ell^1$ norm of a vector, and the matrix norm $|A|_{\mathrm{m}} := \sum_{i=1}^{m} \sum_{j=1}^{n} |A_{ij}|$ for a matrix $A \in \mathbb{R}^{m \times n}$. The errors are measured separately for different parameters $\omega := (\sigma_0, c)$.

The errors in the reconstructed drift function $\hat{\boldsymbol{f}}$ and diffusion function $\hat{\boldsymbol{\sigma}}$ as well as the temporally decoupled squared $W_2$ loss Eq (3) associated with different forms of the diffusion function and different values of $(\sigma_0, c)$ are shown in Fig 4. When the diffusion function is a constant Eq (11), the mean reconstruction error of the drift function is 0.15, the mean reconstruction error of the diffusion function is 0.16, and the mean temporally decoupled squared $W_2$ loss between the ground truth trajectories and the predicted trajectories is 0.074 (averaged over all sets of parameters $(\sigma_0, c)$). When a Langevin-type diffusion function Eq (12) is used as the ground truth, the mean errors for the reconstructed drift and diffusion functions are 0.069 and 0.29, respectively, and the mean temporally decoupled squared $W_2$ loss between the ground truth and predicted trajectories is 0.020. For a linear-type diffusion function as the ground truth, mean reconstruction errors of the drift and diffusion functions are 0.19 and 0.41, respectively, and the mean temporally decoupled squared $W_2$ distance is 0.013. For all three forms of diffusion, our END-nSDE method can accurately reconstruct the drift function $(-\alpha x - \beta y, \beta x - \alpha y)$ (see Fig 4D–4F). When the diffusion function is a constant, our END-nSDE model can also accurately reconstruct this constant (see Fig 4G). When the diffusion function takes a more complicated form such as the Langevin-type diffusion function Eq (12) or the linear-type diffusion function Eq (13), the reconstructed nSDE model can still approximate the diffusion function well for most combinations of $(\sigma_0, c)$, especially when the correlation $c > 0.2$ (see Fig 4H–4I). Overall, our proposed END-nSDE model can accurately reconstruct the minimal stochastic circadian dynamical model Eq (9) in the presence of extrinsic noise (different values of $(\sigma_0, c)$); the accuracy of the reconstructed drift and diffusion functions is maintained for most combinations of $(\sigma_0, c)$. While the drift function is reconstructed with high accuracy, the reconstructed diffusion function exhibits larger relative errors, particularly for models with more complex diffusion forms. How errors depend on the functional forms of the diffusion should be investigated.

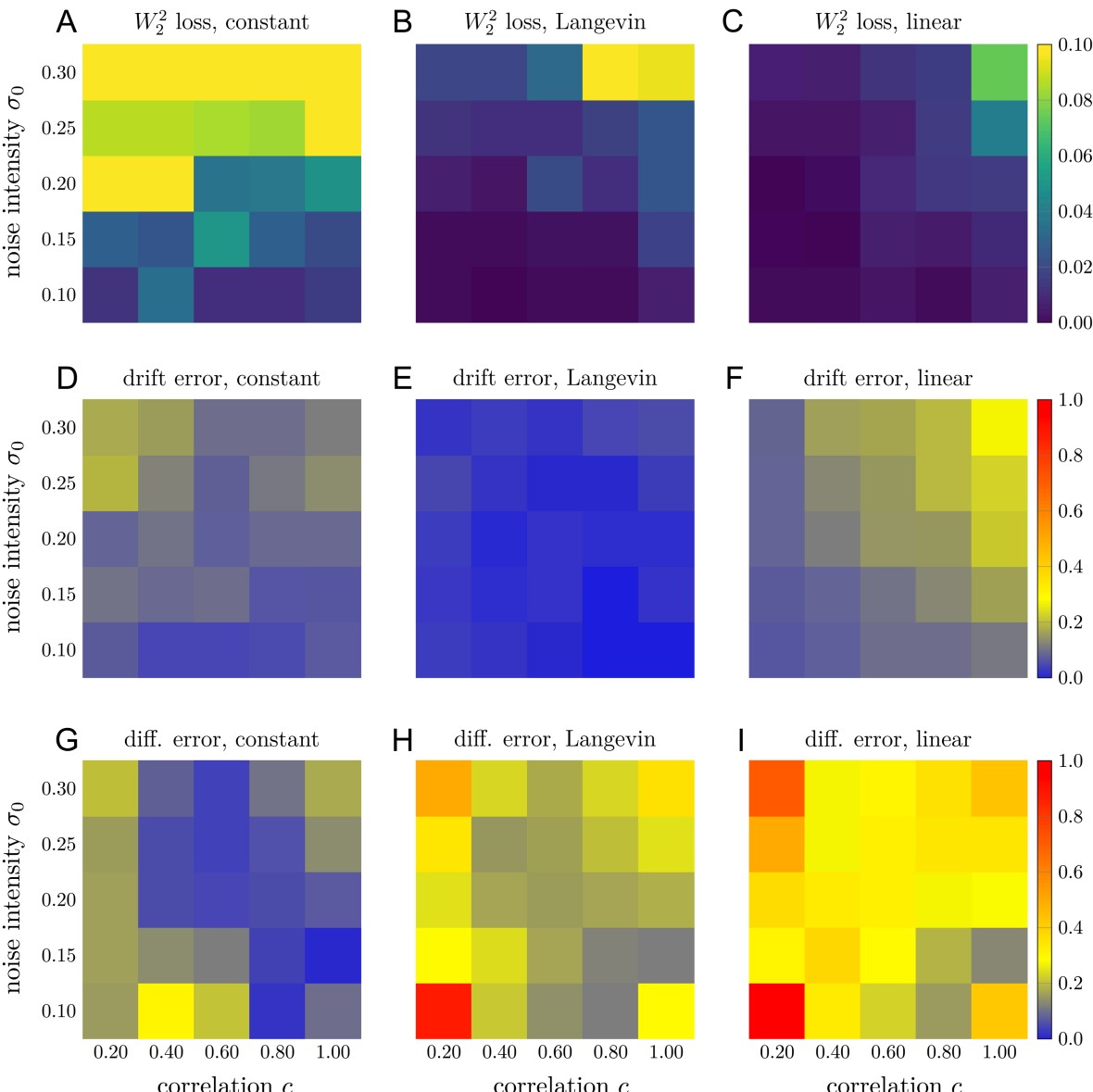

**Fig 4. Reconstructing the circadian model using END-nSDE.** Temporally decoupled squared $W_2$ losses Eq (3) and errors in the reconstructed drift and diffusion functions for different types of the diffusion function and different values of $(\sigma_0, c)$. A-C. The temporally decoupled squared $W_2$ loss between the ground truth trajectories and the trajectories generated by the reconstructed nSDEs for the constant-type diffusion function Eq (11), Langevin-type diffusion function Eq (12), and the linear-type diffusion function Eq (13). D-F. Errors in the reconstructed drift function for the three different types of ground truth diffusion functions and the linear-type diffusion function Eq (13). G-I. Errors in the reconstructed diffusion function for the three different types of ground truth diffusion functions.

To investigate how the strengths of the extrinsic and intrinsic noise and affect our reconstruction of extrinsic-noise-driven SDEs, we conduct an additional test on the reconstruction of circadian clock dynamics. We generate training trajectories from a revised version of Eq (9):

$$
\begin{aligned}
dx &= -\alpha x dt - \beta y dt + (\sigma_0 + \sigma_1 k_1)\sqrt{x} dB_{1,t} \\
dy &= \beta x dt - \alpha y dt + (\sigma_0 + \sigma_1 k_2)\sqrt{y} dB_{2,t}.
\end{aligned}
\tag{16}
$$

In Eq (16), for each set of $(\sigma_0, \sigma_1)$, we generate 25 groups of $(k_1, k_2) \in \{0, \pm0.5, \pm1\} \times \{0, \pm0.5, \pm1\}$ with each group containing 50 trajectories as training data. To train the neural SDE model, both the state variables $(x,y)$ and $(k_1, k_2)$ are input into the neural SDE. $\sigma_0$ characterizes the average level of intrinsic noise while $\sigma_1$ represents the strength of extrinsic noise, and we use different values of $(\sigma_0, \sigma_1)$. As shown in Fig 5B and 5C, errors in the reconstructed drift function and in the reconstructed diffusion function, averaged over all different sets of $(k_1, k_2)$, increases with both $\sigma_0$ and $\sigma_1$. Specifically, an increase in the intrinsic noise level $(\sigma_0)$ reduces the reconstruction accuracy more than an increase in the extrinsic noise $(\sigma_1)$ does. More analysis on how the variation in intrinsic noise and extrinsic noise could affect the accuracy of the reconstructed drift and diffusion functions using our proposed END-nSDE method is promising.

## 3.2. Accurate approximation of interacting DNA-protein systems with different kinetic parameters

To construct a differentiable surrogate for stochastic simulation algorithms (SSAs), the neural SDE model should be able to take kinetic parameters as additional inputs. Thus, the original $W_2$-distance SDE reconstruction method in [32] can no longer be applied because the trained neural SDE model cannot take into account extrinsic noise [60], *i.e.*, different values of kinetic parameters. To be specific, we vary one parameter (the conversion rate $k_2$ from 20nt-mode RPA to 30nt-mode RPA) in the stochastic model and then apply our END-nSDE method which takes the state variables and the kinetic parameter $k_2$ as the input. We set $k_2 \in \{10^{-4+j/10}, j = 0, ..., 25\}$ with other parameters taken from experiments [47] ($k_1 = 10^{-3}$ s$^{-1}$, $k_{-1} = 10^{-6}$ s$^{-1}$, $k_{-2} = 10^{-6}$ s$^{-1}$, see Fig 2). For each $k_2$, we generate 100 trajectories and use 50 for the training set and the other 50 for the testing set. Each trajectory encodes the dynamics of the fraction of 20nt-mode DNA-bound RPA $x_1(t)$ and the fraction of 30nt-mode DNA-bound RPA $x_2(t)$.

When approximating the dynamics underlying the RPA-DNA binding process, we compare our SDE reconstruction method with other benchmark time-series analysis or reconstruction approaches, including the RNN, LSTM, Gaussian process, and the neural ODE model [61,62]. These benchmarks are described in detail in Appendix C in S1 Text.

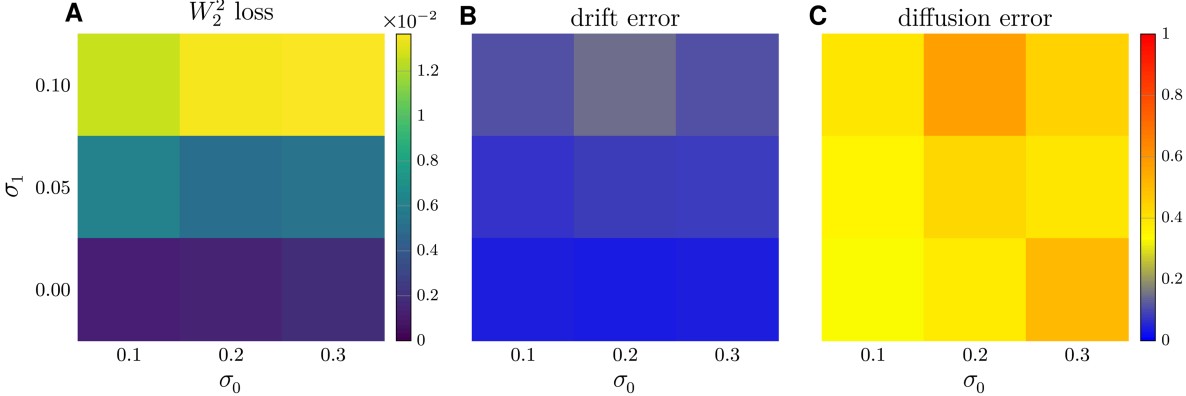

**Fig 5.** Average temporally decoupled squared $W_2$ losses Eq (3) and errors in the reconstructed drift and diffusion functions for different choices of intrinsic noise strength and extrinsic noise strength $(\sigma_0, \sigma_1)$ in Eq (16).

The extrinsic-noise-driven temporally decoupled squared $W_2$ distance loss Eq (8) between the distribution of the ground truth trajectories and the distribution of the predicted trajectories generated by our END-nSDE reconstructed SDE model is the smallest among all methods (shown in Table 1). The underlying reason is an SDE well approximates the genuine Markov counting process underlying the continuum-limit RPA-DNA binding process [49]. The RNN and LSTM models do not capture the intrinsic fluctuations in the counting process. The neural ODE model is a deterministic model and cannot capture the stochasticity in the RPA-DNA binding dynamics. Additionally, the Gaussian process can only accurately approximate linear SDEs, which is not an appropriate form for an SDE describing the RPA-DNA binding process.

In Fig 6A and 6B, we plot the predicted trajectories obtained by the trained neural SDE model for two different values $\lg k_2 = -4$ and $\lg k_2 = -1.5$. Actually, for all different values of $k_2$, trajectories generated by our END-nSDE method match well with the ground truth trajectories on the testing set, as the temporally decoupled squared $W_2$ loss is maintained small for all $k_2$ (shown in Fig 6C). This demonstrates the ability of our method to capture the dependence of the stochastic dynamics on biochemical kinetic parameters.

**Table 1.** The extrinsic-noise-driven time-decoupled squared $W_2$ distance Eq (8) between the ground truth and predicted trajectories generated by different models on the testing set.

| Model | Loss |
|---|---|
| **END-nSDE** | **0.0006** |
| LSTM | 0.062 |
| RNN | 0.087 |
| nODE | 0.0012 |
| Gaussian Process | 0.0010 |

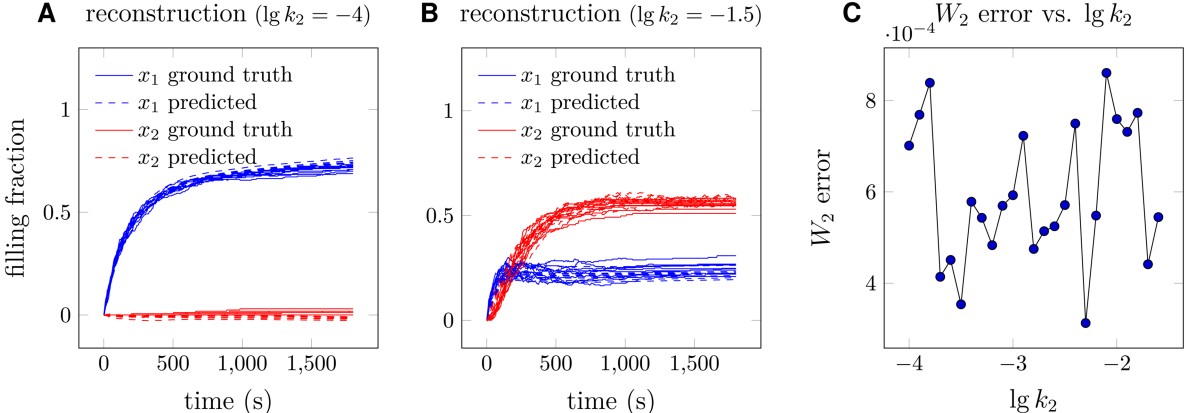

**Fig 6. Reconstructed trajectories of the RPA-DNA binding model.** A. Sample ground truth and reconstructed trajectories evaluated at $\lg k_2 = -4$, where we use the convention that $\lg = \log_{10}$. B. Sample ground truth and reconstructed parameters evaluated at $\lg k_2 = -1.5$. C. Temporally decoupled squared $W_2$ distances (see Eq (8)) between the ground truth and reconstructed trajectories evaluated at different $\lg k_2$ values. In A and B, blue and red trajectories represent the filling fractions of DNA by 20nt-mode and 30nt-mode RPA, respectively. The dashed lines represent the predicted trajectories, and the solid lines represent the ground truth. Throughout the figure, the data are generated by a single neural SDE model that accepts the conversion rate $k_2$ as a parameter and outputs the trajectories.

### 3.3. Reconstructing high-dimensional NF$\kappa$B signaling dynamics from simulated and experimental data

Finally, we evaluate the effectiveness of the END-nSDE framework in reconstructing high-dimensional NF$\kappa$B signaling dynamics under varying noise intensities and investigate the performance of the neural SDE method in reconstructing experimentally measured noisy NF$\kappa$B dynamics. The procedure is divided into two parts. First, we trained and tested our END-nSDE method on synthetic data generated by the NF$\kappa$B SDE model Eq (10) under different noise intensities ($\sigma_1, \sigma_2$). Second, we test whether the trained END-nSDE can reproduce the experimental dynamic trajectories.

**3.3.1. Reconstructing a 52-dimensional stochastic model for NF$\kappa$B dynamics.** For training END-nSDE models, we first generated synthetic data from the 52-dimensional SDE model of NF$\kappa$B signaling dynamics Eqs (10) and established models [51,56]. The synthetic trajectories are generated under 121 combinations of noise intensity ($\sigma_1, \sigma_2$) in Eqs (10) (see Appendix D of S1 Text). The resulting NF$\kappa$B trajectories vary depending on noise intensity, with low-intensity noise producing more consistent dynamics across cells (see Fig 7A) and higher-intensity noise yielding more heterogeneous dynamics (see Fig 7B). The simulated ground truth trajectories are split into training and testing datasets (see Appendix E in S1 Text for details). Specifically, we excluded 25 combinations of noise intensities ($\sigma_1, \sigma_2$) from the training set in order to test the generalizability of the trained neural SDE model on noisy intensities.

Next, as detailed in Appendix E of S1 Text, we trained a 52-dimensional neural SDE model using our END-nSDE method on synthetic trajectories. The loss function is based on the $W_2$ distance between the distributions of the neural SDE predictions in Eqs (10) and the simulated nuclear I$\kappa$B$\alpha$-NF$\kappa$B complex and nuclear NF$\kappa$B activities ($u_5(t)$ and $u_{10}(t)$, respectively) and the corresponding END-nSDE predictions. The remaining 50 variables of the NF$\kappa$B system were treated as latent variables, as they are not directly included in the loss function calculation.

Although the NF$\kappa$B dynamics vary under different noise intensities ($\sigma_1, \sigma_2$), the trajectories generated by our trained neural SDE closely align with the ground truth synthetic NF$\kappa$B dynamics under different noise intensities ($\sigma_1, \sigma_2$) (see Fig 7C and 7D). The neural SDE model demonstrates greater accuracy in reconstructing NF$\kappa$B dynamics when the noise in I$\kappa$B$\alpha$ transcription ($\sigma_1$) is smaller, as evidenced by the reduced squared $W_2$ distance between the predicted and ground-truth trajectories on both the training and validation sets (see Fig 7E and 7F). The temporally decoupled squared $W_2$ loss Eq (8) on the validation set is close to that on the training set for different values of noise intensities ($\sigma_1, \sigma_2$). The mean squared $W_2$ distance across all combinations of noise intensities ($\sigma_1, \sigma_2$) is 0.0013 for the training set, and the validation set shows a mean squared $W_2$ distance of 0.0017.

Since the loss function for this application involves only two variables out of 52, we also tested whether the "full" 52-dimensional NF$\kappa$B system can be effectively modeled by a two-dimensional neural SDE. After training, we found that the reduced model was insufficient for reconstructing the full 52-dimensional dynamics, as it disregarded the 50 latent variables not included in the loss function (see Fig D in Appendix F of S1 Text). This result underscores the importance of incorporating latent variables from the system, even when they are not explicitly included in the loss function.

**3.3.2. Reproducing NF$\kappa$B data with a trained END-nSDE.** We assessed whether our proposed END-nSDE can accurately reconstruct the experimentally measured NF$\kappa$B dynamic

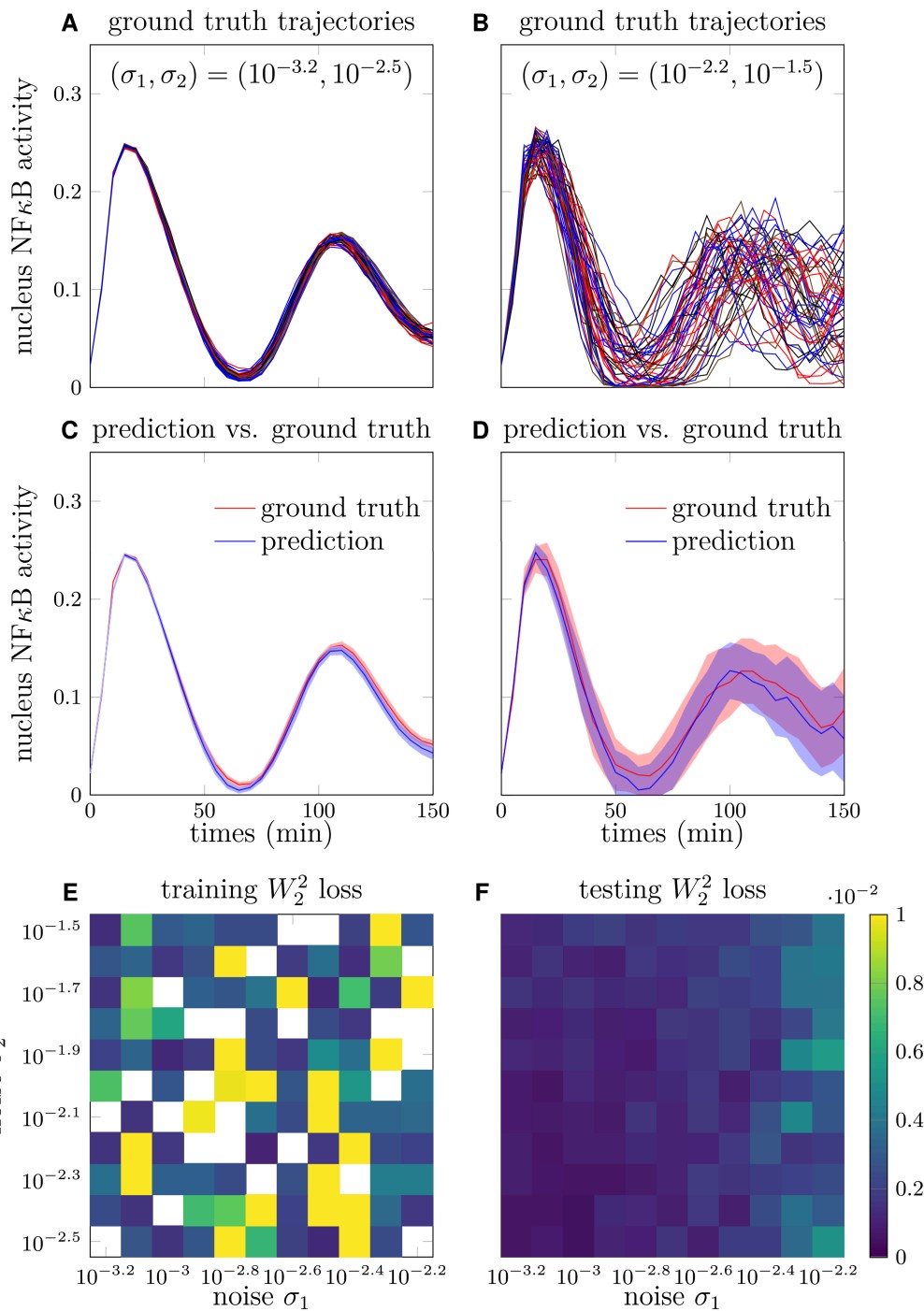

**Fig 7. Reconstruction of NFκB signaling dynamics.** A. Sample trajectories of nuclear NFκB concentration as a function of time with $\sigma_1 = 10^{-3.2}$, $\sigma_2 = 10^{-2.5}$. B. Sample trajectories of nuclear NFκB concentration as a function of time with $\sigma_1 = 10^{-2.2}$, $\sigma_2 = 10^{-1.5}$. C. Reconstructed nuclear NFκB trajectories generated by the trained neural SDE versus the ground truth nuclear NFκB trajectories under noise intensities $\sigma_1 = 10^{-3.2}$, $\sigma_2 = 10^{-2.5}$ in Eq (10). D. Reconstructed nuclear NFκB trajectories generated by the trained neural SDE versus the ground truth nuclear NFκB trajectories under noise intensities $\sigma_1 = 10^{-2.2}$, $\sigma_2 = 10^{-1.5}$. E. The squared $W_2$ distance between the distributions of the predicted trajectories and ground truth trajectories on the training set under different noise strengths $(\sigma_1, \sigma_2)$. For training, we randomly selected 50% sample trajectories in 80 combinations of noise strengths $(\sigma_1, \sigma_2)$ as the training dataset. Blank cells indicate that the corresponding parameter set is not included in the training set. F. Validation of the trained model by evaluating the squared $W_2$ distance between the distributions of predicted trajectories and ground truth trajectories on the validation set.

trajectories. For simplicity and feasibility, we tested the END-nSDE under the assumption that: (1) all cells share the same drift function, and (2) cells with trajectories deviating similarly from their ODE predictions have the same noise intensities. Based on these assumptions, we developed the following workflow (see Fig 8):

1. We used experimentally measured single-cell trajectories of NF$\kappa$B concentration, obtained through live-cell image tracking of macrophages from mVenus-tagged RelA mouse with a frame frequency of five minutes [63], yielding a total of 31 consecutive time points. These trajectories correspond to the sum of nuclear I$\kappa$B$\alpha$–NF$\kappa$B and NF$\kappa$B concentration in the 52D SDE model ($u_5(t)$ and $u_{10}(t)$ in Eq (10)).

2. The experimental dataset was divided into subgroups. Cosine similarity was calculated between the ODE-generated trajectory (representative-cell NF$\kappa$B dynamics) and experimental trajectories. The trajectories are then ranked and divided into different groups based on their cosine similarity with the trajectory generated from the ODE model [64]. Experimental trajectories with higher similarity to the ODE trajectory are expected to exhibit smaller intrinsic fluctuations, corresponding to lower noise intensities (see Appendix G in S1 Text for details).

3. Each group of experimental trajectories was input into the trained neural network (see the next paragraph for more details) to infer the corresponding noise intensities ($\sigma_1, \sigma_2$). For simplicity, we assume that trajectories within each group share the same noise intensities.

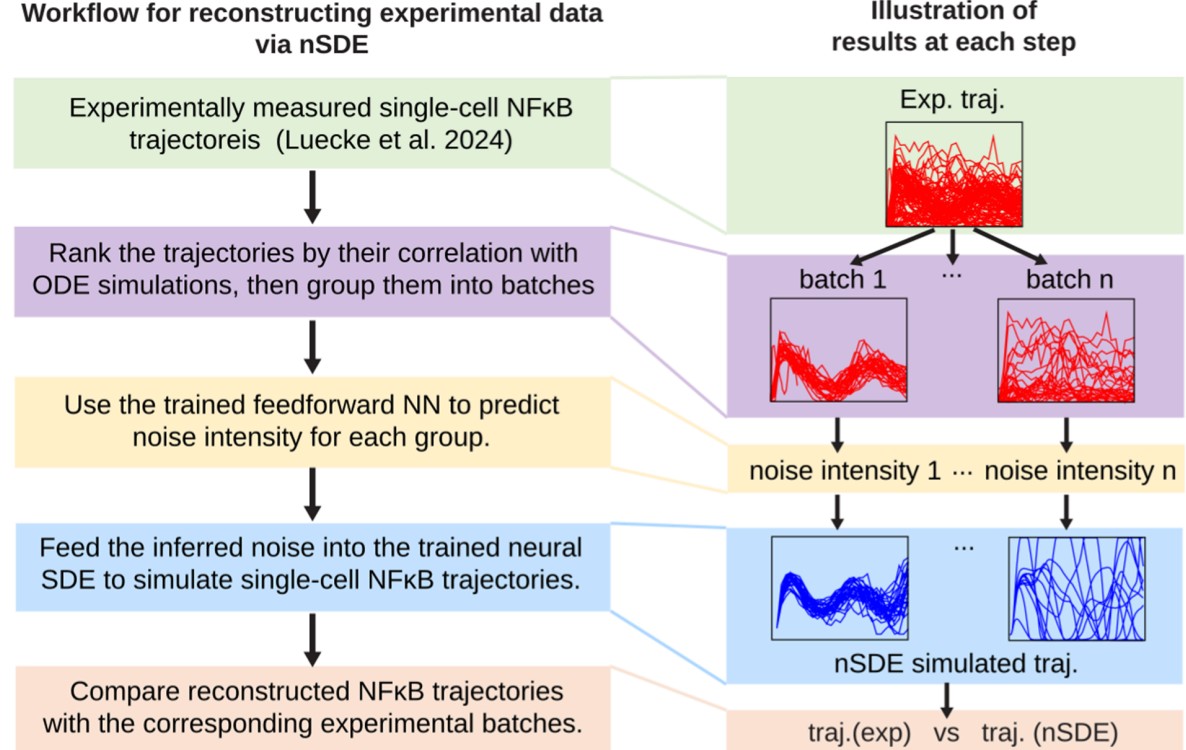

**Fig 8. Workflow of reconstructing experimental data via END-nSDE.** Workflow for reconstructing experimental data using the trained parameterized nSDE and the parameter-inference neural network (NN). The boxes on the left outline the steps of the experimental data reconstruction process, while the boxes on the right illustrate the corresponding results at each step.

4. The inferred noise is then used as inputs for the trained END-nSDE to simulate NFκB trajectories.

5. The simulated trajectories were compared with the corresponding experimental data to evaluate the model's performance.

To estimate noise intensities from different groups of experimentally measured single-cell nuclear NFκB trajectories (step (3) in the proposed workflow), we trained another neural network to predict the corresponding IκBα transcription and NFκB translocation noise intensities from the groups of NFκB trajectories in the synthetic training data, similar to the approach taken in [65]. The trained neural network can then be used for predicting noise intensities in the validation set (see Appendix H in S1 Text for technical details).

Assessing the impact of group size (number of trajectories) on noise intensity prediction performance, we found that taking a group size of at least two leads to a relative error of around 0.1 (see Fig 9A). Given the high heterogeneity present in experimental data, we took a group size of 32 as the input into the neural network. Under this group size, the relative errors in the predicted noise intensities were 0.021 on the training set and 0.062 on the testing set (see Fig 9B and 9C).

Using the trained neural network, we inferred noise intensities for the experimental data, which were grouped based on their cosine similarities with the representative-cell trajectory (deterministic ODE) with a group size of 32. The predicted noise intensities on the experimental dataset are larger than the noise intensities on the training set, possibly because unmodeled extrinsic noise complicates the inference of noise intensity. The transcription noise of IκBα is predicted to be within the range of $[10^{-0.81}, 10^{-0.71}]$ (see Fig 9D). In addition, the inferred noise for NFκB translocation fell within $[10^{-0.49}, 10^{-0.43}]$ (see Fig 9D). These inferred noise intensities were then used as inputs to the END-nSDE to simulate NFκB trajectories.

We compare the reconstructed NFκB trajectories generated by the trained neural SDE model with the experimentally measured NFκB trajectories (see Fig 9E–9I). The trajectories generated using our END-nSDE method successfully reproduce the experimental dynamics for the majority of time points for the top 50% of cell subgroups most correlated with the representative-cell ODE model (see Fig 9E–9G, Fig 9I).

For the top-ranked subgroups (#1 to #16), the heterogeneous nSDE-reconstructed dynamics align well with the experimental data for the first 100 minutes. The predicted trajectories deviate more from ground truth trajectories observed in experiments after 100 minutes possibly due to error accumulation and errors in the predicted noise intensity. For experimental subgroups that significantly deviate from the representative-cell ODE model, the END-nSDE struggles to fully capture the heterogeneous trajectories. This limitation likely arises from the assumption that all cells in a group share the same underlying dynamics, whereas in reality, substantial cellular differences in underlying dynamics exist due to heterogeneity in the drift term, an aspect not accounted for in END-nSDE due to the high computational cost.

With sufficient data and computational resources, our proposed workflow is able to incorporate extrinsic noise in the drift terms, allowing for further discrimination of experimental trajectories. Our END-nSDE method can partially reconstruct experimental datasets and has the potential to fully capture experimental dynamics. Furthermore, trajectories generated from the trained END-nSDE model can reproduce the intrinsic fluctuations in the observed NFκB signaling dynamics which are inaccessible to the representative-cell ODE model.

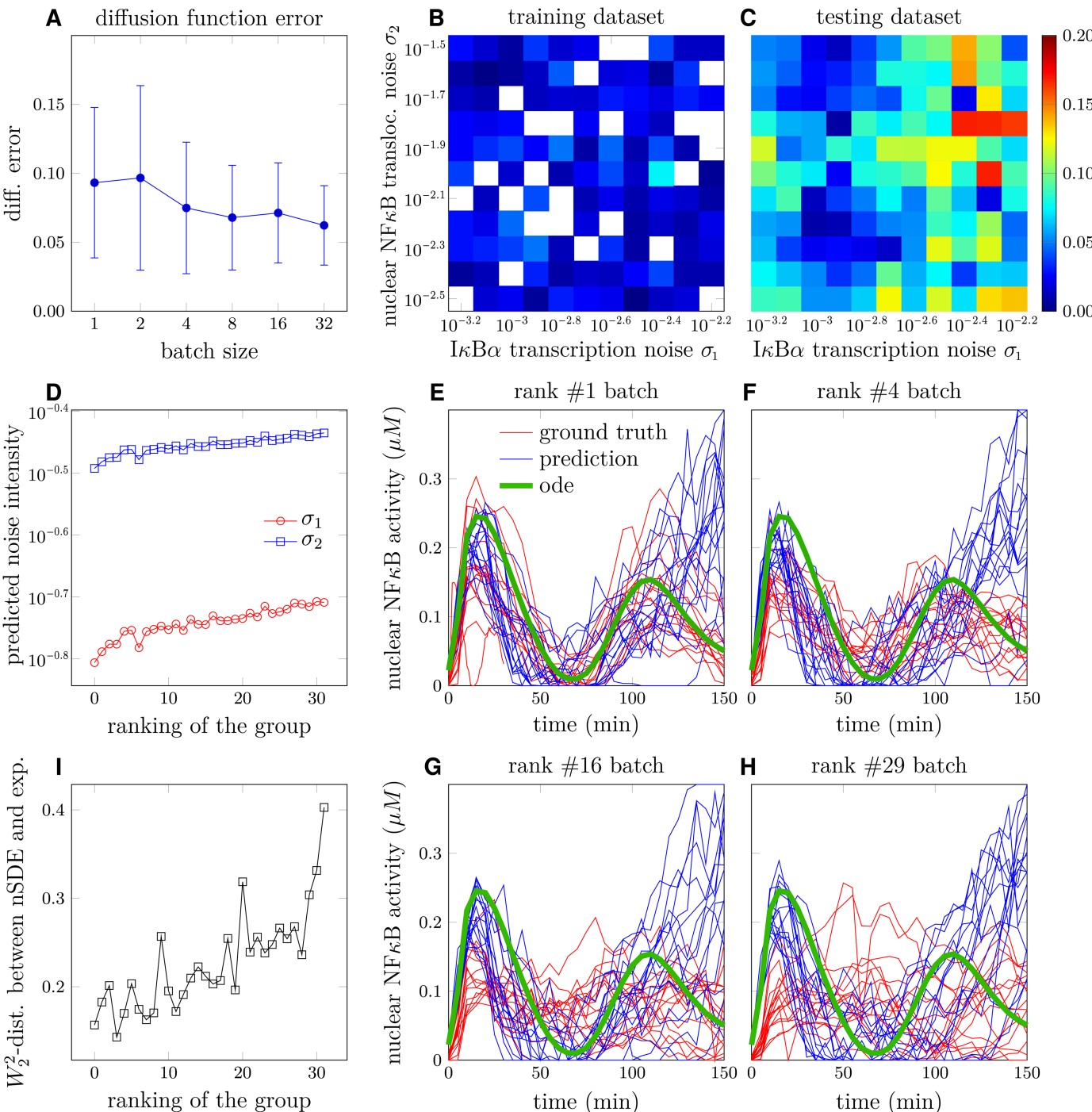

**Fig 9. Inferring intrinsic noise intensities and reconstructing experimental data via END-nSDE.** A. Plots showing the mean (solid circles) and variance (error bars) of the relative error in the reconstructed noise intensities ($\hat{\sigma}_1, \hat{\sigma}_2$) predicted by the parameter-inference NN for the testing dataset, as a function of the group size of input trajectories. B. Heatmaps showing the relative error in the reconstructed noise intensities for the training dataset. Colored cells represent results from the parameter-inference NN for the training dataset, while blank cells indicate noise strength values not included in the training set. C. Heatmaps showing the relative error in the diffusion function for the testing dataset. D. The inferred intensity of IκBα transcription noise ($\sigma_1$) and NFκB translocation noise ($\sigma_2$) in different groups of experimental trajectories, plotted against the group's ranking in decreasing similarity with the representative ODE trajectory. E-H. Groups of experimental and nSDE-reconstructed trajectories ranked by decreasing cosine similarity: #1 (E), #4 (F), #16 (G), #29 (H). The squared $W_2$-distance between experimental and SDE-generated trajectories are 0.157 (E), 0.143 (F), 0.212 (G), 0.236 (H). The inferred noises are ($10^{-0.49}, 10^{-0.81}$) (E), ($10^{-0.47}, 10^{-0.78}$) (F), ($10^{-0.46}, 10^{-0.74}$) (G), ($10^{-0.44}, 10^{-0.71}$) (H). I. The temporally decoupled squared $W_2$ distance between reconstructed trajectories generated by the trained END-nSDE and groups of experimental trajectories, ordered according to decreasing similarity with the representative ODE trajectory.

## 4. Discussion

In this work, we used the $W_2$-distance to develop an END-nSDE reconstruction method that takes into account extrinsic noise in gene expression dynamics as observed across various biophysical and biochemical processes such as circadian rhythms, RPA-DNA binding, and NF$\kappa$B translocation. We first demonstrated that our END-nSDE method can successfully reconstruct a minimal noise-driven fluctuating SDE characterizing the circadian rhythm, showcasing its effectiveness in reconstructing SDE models that contain both intrinsic and extrinsic noise. Next, we used our END-nSDE method to learn a surrogate extrinsic-noise-driven neural SDE, which approximates the RPA-DNA binding process. Molecular binding processes are usually modeled by a Markov counting process and simulated using Monte-Carlo-type stochastic simulation algorithms (SSAs) [48]. Our END-nSDE reconstruction approach can effectively reconstruct the stochastic dynamics of the RPA-ssDNA binding process while also taking into account extrinsic noise (heterogeneity in biological parameters among different cells). Our END-nSDE method outperforms several benchmark methods such as LSTMs, RNNs, neural ODEs, and Gaussian processes.

Finally, we applied our methodology to analyze NF$\kappa$B trajectories collected from over a thousand cells. Not only did the neural SDE model trained on the synthetic dataset perform well on the validation set, but it also partially recapitulated experimental trajectories of NF$\kappa$B abundances, particularly for subgroups with dynamics similar to those of the representative cell. These results underscore the potential of neural SDEs in modeling and understanding the role of intrinsic noise in complex cellular signaling systems [66–68].

When the experimental trajectories were divided into subgroups, we assumed that all cells across different groups shared the same drift function (as in the representative ODE) and cells within each group shared the same diffusion term. We found that subgroups with dynamics more closely aligned with the deterministic ODE model resulted in better reconstructions. In contrast, for experimental trajectories that deviated significantly from the representative ODE model, their underlying dynamics may differ from those defined by the representative cell's ODE. Therefore, the assumption that a group shares the same drift function as the representative cell ODE holds only when the trajectories closely resemble the ODE. Incorporating noise into the drift term for training the neural SDE could potentially address this issue. We did not consider this approach due to the high computational cost required for training.

Applying our method to high-dimensional synthetic NF$\kappa$B datasets, we showed the importance of incorporating latent variables. This necessity arises because the ground-truth dynamics of the measured quantities (nuclear NF$\kappa$B) are not self-closed and inherently depend on additional variables. Consequently, the 52-dimensional SDE reconstruction requires more variables than just the "observed" dynamics of nuclear NF$\kappa$B. In this example, the remaining 50 variables in the nSDE were treated as latent variables, even though they were not explicitly included in the loss function.

For high-dimensional systems (*e.g.*, 52 dimensions as in our NF$\kappa$B example), analyzing stochastic dynamics remains challenging. Even though regulated processes do not follow gradient dynamics in general, imposing a self-consistent energy landscape and adopting lower dimensional projections can provide a valuable framework for studying stochastic dynamics of high-dimensional biological systems [69–71]. Once an effective energy landscape is identified, prior knowledge about the system structure can be incorporated into the neural SDE framework through the following formulation:

$$d\boldsymbol{X} = \big(F(\boldsymbol{X}) + \hat{\boldsymbol{f}}(\boldsymbol{X}; \omega)\big)dt + \hat{\boldsymbol{\sigma}}(\boldsymbol{X}; \omega)d\boldsymbol{B}_t, \tag{17}$$

where $F(\boldsymbol{X}) = \nabla E(\boldsymbol{X})$ represents the prior knowledge of the energy landscape. The neural networks $\hat{f}$ and $\hat{\sigma}$ then learn deviations from the prior knowledge and the unknown intrinsic noise. Such prior information on the energy landscape could facilitate training and improve accuracy of the learned model [37]. How imposition of a high-dimensional landscape as a constraint affects our $W_2$-distance-based inference and how this potential sold be interpreted should be explored in more depth. If meaningful and informative, prior results on how landscapes can be used to characterize neural networks across various tasks can be leveraged [72, 73].

Finally, neural SDEs can serve as surrogate models for complex biomedical dynamics [74, 75]. Combining such surrogate models with neural control functions [72,76,77] can be useful for tackling complex biomedical control problems. As shown in preceding work [32,37], a larger number of training trajectories led to a more accurate reconstructed neural SDE. However, in biological experiments, obtaining more training trajectories could be more expensive. Therefore, it is of biological significance to find out the number of training trajectories that can be practically obtained in real experiments and that are necessary for an accurate reconstruction of the intrinsic-noise-aware SDE using our END-nSDE approach. Finally, it is worth further investigation to find out the biophysical molecular processes in which taking into account intrinsic fluctuations is necessary. In such problems, using our END-nSDE framework to reconstruct the noisy molecular dynamics could yield a more biologically reasonable, noise-aware model than first-principle-based mass-action ODE models.

While our work focuses on gene regulation dynamics, it is important to emphasize that the END-nSDE reconstruction method is general and can potentially be applied to biological systems beyond gene regulation. The method's ability to capture both intrinsic and extrinsic noise makes it suitable for modeling various stochastic biological processes, including, but not limited to, signal transduction networks, metabolic pathways, population dynamics, and developmental processes. The examples we chose-circadian rhythms, RPA-DNA binding dynamics, and NF$\kappa$B signaling-were selected to demonstrate the method's capabilities and bring the neural SDE approach to the attention of the molecular and cell biology community. Future applications could extend to other domains such as epidemiology, ecology, and systems biology, where stochastic dynamics that could be described by SDEs with heterogeneity among different cells or individuals are prevalent.

Besides better understanding effective energy landscape constraints, there are several promising directions for future research. First, techniques to extract an explicit form of the learned neural network SDEs can be developed. For example, one could employ a polynomial model as the reconstructed drift and diffusion functions in the SDE [78]. Such an explicit functional form of the approximate SDE may facilitate biological interpretation of the underlying model. Recent research has also shed light on directly interpreting trained neural networks using simple functions such as polynomials [79]. Therefore, one can apply such methods to extract the approximate forms from the learned drift and diffusion functions in the neural SDE and interpret their biophysical meaning.

Another promising avenue of investigation is to combine discrete and continuous modeling approaches to account for both mRNA and protein dynamics. Such a hybrid approach would use discrete Markov jump processes for low-abundance species (such as mRNA) while employing SDEs for high-abundance species (such as proteins), thereby addressing the limitations of pure SDE approaches when molecular counts approach zero.

Finally, the presence of unobserved variables in cellular systems poses a significant challenge for accurate SDE modeling. Many cellular processes involve hidden regulatory mechanisms, unmeasured metabolites, or latent cellular states that influence the observed dynamics but are not directly captured in experimental measurements. This limitation can lead to

model misspecification, where the inferred drift and diffusion functions compensate for missing variables, potentially resulting in biased parameter estimates and poor predictive performance. A more realistic scenario occurs when we already know what molecules can have an effect on the dynamics, but experiments can only report a few molecular species. In such cases, we can model the full system dynamics with the full dimension with a parameterized model for sampling the initial values of those unobserved variables. The rest of the training procedure would be the same as in the main text.

## Supporting information

**S1 Text. Technical appendices.**
(PDF)

## Acknowledgments

We acknowledge Stefanie Luecke for providing the experimental datasets for NF$\kappa$B dynamics.

## Author contributions

**Formal analysis:** Xiangting Li, Mingtao Xia.

**Funding acquisition:** Lucas Böttcher, Tom Chou.

**Investigation:** Jiancheng Zhang, Xiangting Li, Xiaolu Guo, Zhaoyi You, Lucas Böttcher, Alex Mogilner, Alexander Hoffmann, Tom Chou, Mingtao Xia.

**Methodology:** Jiancheng Zhang, Xiangting Li, Mingtao Xia, Xiaolu Guo.

**Project administration:** Mingtao Xia.

**Resources:** Xiangting Li, Xiaolu Guo, Lucas Böttcher, Mingtao Xia.

**Software:** Jiancheng Zhang, Xiangting Li, Tom Chou, Mingtao Xia, Xiaolu Guo.

**Supervision:** Xiaolu Guo, Mingtao Xia.

**Validation:** Jiancheng Zhang, Xiangting Li, Mingtao Xia, Xiaolu Guo.

**Visualization:** Jiancheng Zhang, Xiangting Li, Xiaolu Guo, Tom Chou, Mingtao Xia.

**Writing – original draft:** Jiancheng Zhang, Xiangting Li, Xiaolu Guo, Zhaoyi You, Mingtao Xia.

**Writing – review & editing:** Jiancheng Zhang, Xiangting Li, Xiaolu Guo, Zhaoyi You, Lucas Böttcher, Alex Mogilner, Alexander Hoffmann, Tom Chou, Mingtao Xia.

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
