## [Decision Letter · Decision Letter 0]

7 Jun 2025

PCOMPBIOL-D-25-00636

Reconstructing noisy gene regulation dynamics using extrinsic-noise-driven neural stochastic differential equations

PLOS Computational Biology

Dear Dr. Xia,

Thank you for submitting your manuscript to PLOS Computational Biology. After careful consideration, we feel that it has merit but does not fully meet PLOS Computational Biology's publication criteria as it currently stands. Therefore, we invite you to submit a revised version of the manuscript that addresses the points raised during the review process.

As you will see from the attached reviews, several reviewers found the work of interest, but raised significant concerns about the value of modeling stochastic gene expression with SDEs, specifically, what biological insights are gained from estimating the drift and diffusion terms, and how accurately do such terms need to reflect the underlying stochastic distributions to be confident of such inferences. We would be happy to reconsider the manuscript if revisions can be made to address these concerns.

Please submit your revised manuscript within 60 days Aug 07 2025 11:59PM. If you will need more time than this to complete your revisions, please reply to this message or contact the journal office at ploscompbiol@plos.org. Please include the following items when submitting your revised manuscript:

We look forward to receiving your revised manuscript.

Kind regards,

Michael A Beer

Academic Editor

PLOS Computational Biology

Padmini Rangamani

Section Editor

PLOS Computational Biology

**Journal Requirements:**

At this stage, the following Authors/Authors require contributions: Jiancheng Zhang, Xiangting Li, Xiaolu Guo, Zhaoyi You, Lucas Böttcher, Alex Mogilner, Alexander Hoffmann, Tom Chou, and Mingtao Xia. Please ensure that the full contributions of each author are acknowledged in the "Add/Edit/Remove Authors" section of our submission form.

4) We notice that your supplementary Figures are included in the manuscript file. Please remove them and upload them with the file type 'Supporting Information'. Please ensure that each Supporting Information file has a legend listed in the manuscript after the references list.

5) We note that your Data Availability Statement is currently as follows: "No data was created in this research. All data used in this research are publicly available and have been properly cited.Authors will make all codes publicly available upon acceptance of this manuscript.". Please confirm at this time whether or not your submission contains all raw data required to replicate the results of your study. Authors must share the “minimal data set” for their submission. PLOS defines the minimal data set to consist of the data required to replicate all study findings reported in the article, as well as related metadata and methods (https://journals.plos.org/plosone/s/data-availability#loc-minimal-data-set-definition).

6) Please provide a detailed Financial Disclosure statement. This is published with the article. It must therefore be completed in full sentences and contain the exact wording you wish to be published.

1) Please clarify all sources of financial support for your study. List the grants, grant numbers, and organizations that funded your study, including funding received from your institution. Please note that suppliers of material support, including research materials, should be recognized in the Acknowledgements section rather than in the Financial Disclosure

2) State the initials, alongside each funding source, of each author to receive each grant. For example: "This work was supported by the National Institutes of Health (####### to AM; ###### to CJ) and the National Science Foundation (###### to AM)."

3) State what role the funders took in the study. If the funders had no role in your study, please state: "The funders had no role in study design, data collection and analysis, decision to publish, or preparation of the manuscript."

4) If any authors received a salary from any of your funders, please state which authors and which funders..

**Reviewers' comments:**

Reviewer's Responses to Questions

**Comments to the Authors:**

Reviewer #1: Effective control of cellular signaling and gene expression is essential for sustaining cell operations, growth, and responses to environmental shifts under fluctuations. In this work, Zhang et al. propose a neural stochastic differential equation framework driven by extrinsic noise named END-nSDE, to learn cell population dynamics from trajectory data. They validated their method in a few simulated datasets, including circadian oscillations, RPA-DNA interactions, and NFκB signaling, and demonstrated that their approach surpasses conventional time-series tools like RNN and LSTM in accuracy. By quantifying cell-to-cell variations directly from data, their model can replicate experimentally observed stochastic dynamics partially. I think this may be an interesting method for simulating biological systems when detailed mechanistic models are missing, enabling insights into noise-driven cellular processes. I have the following comments:

1, It’s interesting to see that the authors can reconstruct temporal experimental data (Fig. 8). However, for a high-dimensional system (e.g., 52 dimensions referred in this work), how to analyze the stochastic dynamics is a paramount question, even if we already have a dynamical model or Neural SED model, as the authors have emphasized the importance of external and intrinsic noise. For example, the energy landscape theory provides a way to study stochastic dynamics of high-dimensional biological systems (Kang and Li, Adv. Sci., 8: 2003133 (2021), Li and Wang, PNAS 111, 14130-14135 (2014)). So, I suggest the authors considering adding related discussions.

2, The authors have used Wasserstein distance. Can it be replaced by KL-divergence? If so, what is the advantage and disadvantage for using these two measures regarding current task?

3, For reconstructing experimental data (Fig. 8), how many time points are there from experiments? And how does the proposed approach depend on the number (and resolution) of input experimental data point? Please elaborate this point.

Reviewer #2: The paper by Zhang et al reports on a novel method that attempts to reconstruct the stochastic dynamics of gene regulatory networks using neural stochastic differential equations (SDEs). Basically the method works by approximating the drift and diffusion functions of the SDEs using neural networks. The reconstructed neural SDE is then in principle a surrogate model of single-cell dynamics. The method is tested on 3 different models of intracellular dynamics. Their claim is that the surrogate model can effectively capture both intrinsic and extrinsic noise. The paper is overall well written and certainly the application of these approaches is promising. I am however less convinced by the claims of its accuracy, the limitations of the method are not sufficiently addressed and the authors seem to have missed a large amount of relevant literature on this topic. If these can be addressed then I think the paper will present a much stronger and thorough exposition of what is an interesting and potentially useful method. Here are my more detailed comments:

1. It is practically immediately assumed that stochastic gene expression can be suffuciently modelled by an effective SDE. I strongly disagree with this statement. SDEs can only be derived from first principles in the large system size limit and gene systems are not in this category because genes and mRNA numbers are typically present in very low copy numbers per cell which automatically means a discrete description in terms of an effective master equation is a much more accurate. Having said this, if the genes and mRNA are not explciitly described in the model, i.e. the model describes protein dynamics (proteins are typcially quite abundant per cell) and the gene regulatory interactions are modelled implicitly by Hill functions and similar then yes an effective SDE approach starts to make sense. To make this point, I note that in mouse cells, it is reported that the median mRNA number is about 17 while the median protein numbers is about 50,000 (Schwanhäusser et al. "Global quantification of mammalian gene expression control." Nature 473.7347 (2011): 337-342). So the first point is to clarify from the beginning which systems can be suitably approximated by an effective SDE approach. Of course, one can try to approximate such systems still with an SDE approach but in that case the learnt drift and diffusion functions will not have much physical meaning. Also I would conjecture that very likely the estimated diffusion function would be quite badly approximated as its very difficult to capture the noise properly using a continuum model when ithe input stochastic process regularly hits zero (would be the case for most mRNA species due to their very low abundance in most cells as shown by sequencing studies which regularly report a mean mRNA is that is between 1 and 5).

2. I note that machine-learning based approaches to learn surrogate discrete models from low abundance gene expression data already exist: Jiang et al. "Neural network aided approximation and parameter inference of non-Markovian models of gene expression." Nature communications 12.1 (2021): 2618; Cao et al. "Efficient and scalable prediction of stochastic reaction–diffusion processes using graph neural networks." Mathematical biosciences 375 (2024): 109248; Öcal et al. "Inference and uncertainty quantification of stochastic gene expression via synthetic models." Journal of The Royal Society Interface 19.192 (2022): 20220153; Sukys et al. "Approximating solutions of the chemical master equation using neural networks." Iscience 25.9 (2022). Some of these approaches are particularly close in spirit to the present approach in the sense that they learn the propensity functions of an effective master equation (the propensity functions of the master equation determine the drift and diffusion functions in an SDE as discussed in the Chemical Langevin equation paper by D. T. Gillespie). The existing literature on approaches that aim to do the same (or similar) but using master equations should be cited and discussed vis-a-vis the nSDE approach.

3. On Section 3.1 on Circadian clock dynamics, it is mentioned "Overall, our proposed END-nSDE model can accurately reconstruct the minimal stochastic circadian dynamical model". I think this is overstating the accuracy of their method and similar claims are made in other parts of the paper. In particular I note that the accuracy is reasonably good for the drift function but not for the diffusion function -- they report relative errors of 0.29 for the Langevin-type diffusion and 0.41 for the linear-type diffusion model which are really high! The drift is comparatively much more accurately reconstructed but this is to be expected because this is the deterministic part of the SDE. I also think they should more carefully assess the accuracy of the nSDE using a more systematic approach, particularly to understand how well the reconstruction of the drift and diffusion functions vary as a function of the size of the intrinsic and extrinsic noise in the data (as measured by the coefficient of variation - CV). I would expect to see the errors increase as the CV increases. So to summarise it is important that the limitations of the method are properly and thoroughly investigated so the reader can understand when the method can be applied and when other approaches maybe more suitable.

Reviewer #3: The authors present a method for reconstructing dynamical system trajectories that involve a combination of intrinsic (random fluctuations in biochemical reactions) and extrinsic (cell-to-cell heterogeneity) noise. Their approach is based on previous Wasserstein distance-based neural stochastic differential equations (SDE) models. They train neural negworks to learn the drift and diffusion functions of the SDEs. The method is used on three biological systems: a trivial (linear) model of damped oscillator that is meant to capture some aspect of circadian rhythms, RPA-DNA binding, and NFκB signaling.

Overall, the paper is interesting. I did not always find it is clear; important steps in the methodology are not well explained and believe that major changes are needed.

1. The authors claim that the major difference between the present manuscript and reference 29 is that the current manuscript is the inclusion of extrinsic noise.

2. The Methods section (2.1) left me confused about the “neural SDE reconstruction method” – two SDEs are presented in equations 1 and 2. Is f-hat the same as f? The dimensionality of the states is the same (real d-vectors). Is this not placing significant constraints, since the dimension of any actual biological system is likely to be unknown and much higher than what the model attempts?

3. The notation for $\pi(\mu(t),\hat{\mu}(t))$ is not very clear as there appears to conflict with the definitions in Equation 4 (in one case the two arguments are probability distributions, in the other there is a Borel \sigma-algebra and a linear space).

4. I did not find the cartoon in Figure 1A,B to be sufficient to understand the END-nSDE method. This appears to be the main contribution of the manuscript, and it would be essentially impossible to recreate the results based on the information provided.

5. Model 1: Circadian clock model. I was somewhat confused by the fact that the paramaters c and \sigma_0 are inputs to the system. Would this not be something that should be found by the methodology?

6. Model 3: Several observations here. On the plus side, the methodology is being used on real data which is great. However, the results leave a few questions. The initial training required a detailed model. To what extent is the END-nSDE and improvement over this existing model?

7. This example suggests that there is a lot of pre-processing before the method is used (e.g. the steps on page 12). How generalizable is this?

8. Last major point. If everything works well, the END-nSDE technique recreates trajectories. It is not clear, however, what the practical benefit of this is. Did we learn something new about the biology of the systems? Not that I can see – the drift and diffusion terms obtained are essentially black boxes with no insight into the biology. I think that the argument is that these models could be used to train controllers of some sort. More discussion on this would be welcome.

9. Minor point: What is the subscript “m” in equation 15?

10. The title includes “gene regulation dynamics” but there is nothing intrinsical about the method that woud apply only to “gene” dynamics. Why not make it more general?

11. Typo: “fluctuations in the single-cell circadian rhythm is [sic] noise-induced” (p. 5)

12. Typo: “Hyperparameters in the neural network is [sic] initialized” p. 22

**Have the authors made all data and (if applicable) computational code underlying the findings in their manuscript fully available?**

Reviewer #1: Yes

Reviewer #2: Yes

Reviewer #3: **No: **Experimental data for NFkB model was not included. Software for methods was not included (although promised following acceptance)

PLOS authors have the option to publish the peer review history of their article (what does this mean?). If published, this will include your full peer review and any attached files.

Reviewer #1: No

Reviewer #2: No

Reviewer #3: No

**Figure resubmission:**
---

## [Decision Letter · Decision Letter 1]

24 Aug 2025

Dear Prof. Xia,

We are pleased to inform you that your manuscript 'Reconstructing noisy gene regulation dynamics using extrinsic-noise-driven neural stochastic differential equations' has been provisionally accepted for publication in PLOS Computational Biology.

Best regards,

Michael A Beer

Academic Editor

PLOS Computational Biology

Padmini Rangamani

Section Editor

PLOS Computational Biology

Reviewer's Responses to Questions

**Comments to the Authors:**

Reviewer #1: The authors have fully addressed my previous comments. I recommend the publication of this paper.

Reviewer #2: The authors have done a substantial revision that addresses all of my comments. I recommend the paper's acceptance in its current form.

Reviewer #3: I appreciate the responses received. As in my previous review, I continue to have concerns about the usefulness of the results, as so much information is needed a priori ("relevant molecular species influencing the dynamics are known a priori", "known, parameterized level of extrinsic noise") leading to a blackbox model. However, I can also appreciate that this is a first step which will likely spur further investigation - as such, I can support publication.

**Have the authors made all data and (if applicable) computational code underlying the findings in their manuscript fully available?**

Reviewer #1: Yes

Reviewer #2: Yes

Reviewer #3: Yes

PLOS authors have the option to publish the peer review history of their article (what does this mean?). If published, this will include your full peer review and any attached files.

Reviewer #1: **Yes: **Chunhe Li

Reviewer #2: No

Reviewer #3: No